# Hair Cortisol, Testosterone, Dehydroepiandrosterone Sulfate and Their Ratios in Stallions as a Retrospective Measure of Hypothalamic–Pituitary–Adrenal and Hypothalamic–Pituitary–Gonadal Axes Activity: Exploring the Influence of Seasonality

**DOI:** 10.3390/ani11082202

**Published:** 2021-07-25

**Authors:** Sergi Olvera-Maneu, Anaïs Carbajal, Jaume Gardela, Manel Lopez-Bejar

**Affiliations:** 1Department of Animal Health and Anatomy, Veterinary Faculty, Universitat Autònoma de Barcelona, Bellaterra (Cerdanyola del Vallès), 08193 Barcelona, Spain; anais.carbajal@uab.cat (A.C.); jaume.gardela@uab.cat (J.G.); 2College of Veterinary Medicine, Western University of Health Sciences, 309 East Second Street, Pomona, CA 91766, USA

**Keywords:** hair, testosterone, cortisol, dehydroepiandrosterone sulphate, ratio, season, horse, stallion, non-invasive, hypothalamic–pituitary–adrenal axis, hypothalamic–pituitary–gonadal axis

## Abstract

**Simple Summary:**

The monitoring of a horse’s endocrine status can provide valuable information of its welfare and sexual and health conditions. The use of non-invasive matrixes for hormonal monitoring, as alternatives to blood samples, are being increasingly used. Among them, the measurement of hormones deposited in the hair shaft has been widely used as a retrospective hormonal biomarker. In the present study, we aimed to evaluate the seasonal variations in hair C, T and DHEA-S in horses for a whole year, as well as to assess the variations between seasons of C/DHEA-S and T/C ratios as a retrospective measure of the hypothalamic–pituitary–adrenal and hypothalamic–pituitary–gonadal axis activity. The results showed how cortisol, testosterone and dehydroepiandrosterone sulphate and their ratios were significantly affected by season. Therefore, as shown in this study, season should be considered when analysing sexual and stress hormones in stallion hair.

**Abstract:**

The monitoring of stress physiology includes studying a wide range of endocrinological mechanisms, which can be assessed using multiple tissue samples. This study aimed to evaluate the seasonal variations of hair C, T and DHEA-S in horses for a whole year, as well as to assess the variations between seasons of C/DHEA-S and T/C ratios as a retrospective measure of the hypothalamic–pituitary–adrenal and hypothalamic–pituitary–gonadal axis activity. Ten pure-breed Menorca stallions were included in the study. The hair samples were collected approximately every two months following the shave-reshave method caudally to the sternum. After a methanol-based extraction, samples were analyzed by enzyme immunoassay for cortisol, testosterone, and dehydroepiandrosterone sulphate. Following our findings, we detected that cortisol, testosterone and dehydroepiandrosterone sulphate were significantly affected by seasonality, with the highest values of cortisol during summer and the lowest values of testosterone during spring. Dehydroepiandrosterone sulphate concentrations were increased in autumn compared to the other studied periods. Additionally, the studied hormone ratios showed variations between seasons. To conclude, season should, therefore, be considered when assessing sexual and stress hormones in stallion hair, since this variable can be a potential influencing factor and led to misinterpretations.

## 1. Introduction

The monitoring of a horse’s endocrine status can provide valuable information of its welfare, sexual and health conditions [1]. Recently, there has been growing attention on those endocrine systems that interfere with each other in complex dynamic ways [2], such as the hypothalamic–pituitary–gonadal (HPG) and hypothalamic–pituitary–adrenal (HPA) axis. Hence, assessing simultaneously hormones that are thought to co-regulate to each other, such as those secreted by HPA and HPG activation, would be postulated as a better approach to physiological stress measurements [2,3].

Cortisol (C), the main glucocorticoid in mammals [4,5], is commonly used as a physiological indicator of stress, since it is secreted after the activation of the HPA axis in response to a situation in which homeostasis is threatened by the actions of various external or internal stressors [5,6]. Additionally, with the activation of the HPA axis, other components, such as dehydroepiandrosterone (DHEA) and its sulfate ester (DHEA-S), are released into bloodstream [3]. Both DHEA and DHEA-S have been characterized as glucocorticoid antagonists, and are considered as indicators of resilience, because of their anti-aging, immune-enhancing and neuroprotective functions [3]. Testosterone (T) is the main sexual steroid in males and has been used as a stress indicator, since this hormone is negatively influenced by stressful living environments, impacting reproductive performance, growth rate, and immunity [7]. One common method to achieve a better approach to stress assessment is to calculate the ratio of various hormones [2]. Different studies have suggested the C/T ratio as a catabolic/anabolic equilibrium during physical fitness since T has primarily anabolic effects and C has catabolic functions [2]. On the other hand, C/DHEA ratio has been used as a more complete picture of the HPA activity and as a potential biomarker tool of resilience and allostatic load in livestock animals, including horses [3,8,9,10].

Steroid hormones, as those concerning this paper, are commonly measured in matrixes that reflect either current or recent circulating hormonal concentrations, such as blood, saliva, urine or faeces [5,11,12]. However, the assessment of hair steroid hormones is being increasingly used in domestic and wild species [13]. Measuring hair T, C, and DHEA-S, allows for the assessment of long-term endocrinological retrospective information and has been used as a compilation of HPG and HPA axis activity, respectively [5]. Steroid hormones, especially lipophilic hormones, are incorporated into the hair shaft through passive diffusion from blood and provides an approach to quantify circulating steroid hormones levels accumulated over the hair growth cycle [14,15]. Another advantage of using hair to measure steroid hormone variations is non-invasive and pain-free sample collection [13,15]. Additionally, hair is a stable substrate with less variability than saliva or feces [5] and it can be stored during long periods at room temperature [15,16]. Therefore, measuring steroid hormones in hair is unaffected by circadian variations in the hormone or by factors inducing short-term hormonal variations [14,15].

Several studies in domestic and wild mammals have revealed different sources of variation in hair steroids levels [13], depending on hair specific characteristics (e.g., body region [17], hair growth phase [18], or colour [19,20]), intrinsic animal-based characteristics (e.g., age [19,21], sex [22,23] or reproductive status [24,25]) and, finally, external factors (e.g., season [26,27]). All of the previously presented components should be considered when using hair to assess hormonal levels as stress indicators. 

Previous studies have examined hair cortisol concentrations (HCC) as a presumptive measure of stress, and demonstrated their seasonal fluctuations in horses [18,22,27,28]. Few studies have been performed using T [11,28,29,30] and, to date, to the best of our knowledge, there is no information published on the seasonal variations in the hair T levels of horses. Although the C/DHEA ratio has been previously described in horse’s hair to evaluate the effect of different housing conditions [8], there is still a lack of information on its biological relevance in the horse. 

In the present study, we aimed to evaluate the seasonal variations in hair C, T and DHEA-S in horses for a whole year, as well as to assess the variations between seasons of C/DHEA-S and T/C ratios as a retrospective measure of the hypothalamic–pituitary–adrenal and hypothalamic–pituitary–gonadal axis activity.

## 2. Materials and Methods

### 2.1. Horses, Housing, Diet, Handling, and Ethical Considerations

Ten pure-breed Menorca stallions, a black coated and autochthonous breed from the island of Menorca (Spain), were included in the study. The animals were aged between eight and twelve years old. All individuals were healthy, and no symptoms of illness were detected from at least one month before the beginning of the study. The body condition score (BCS) was evaluated using the 0 (emaciated) to 5 (extremely fat) BCS scale [31]. For all the stallions evaluated in the present study, BCS was 3 (moderate to good body condition). Additionally, a daily observation of the animals was performed by the owners and a general physical examination was performed the sampling days by a veterinarian (S.O.-M.). Horses were habitually housed in *Centro Ecuestre Equimar* (Maó, Menorca, Spain) in individual indoor stables of 10.5 m^2^ with no cooling or heating elements during the different seasons. No changes in facility and housing conditions were carried out during the whole study. Individuals were fed with a similar commercial horse diet 3 times per day with ad libitum access to hay and running water. The horses from the study were all destinated to perform the same type of equestrian discipline, specifically horses were habitually used to Menorca dressage discipline.

Horses were managed following the principles and guidelines of the Ethics Committee on Animal and Human Experimentation from the Universitat Autònoma de Barcelona (Spain) and following the Directive 2010/63/EU on the protection of animals used for scientific purposes. No other manipulation, different to shaving a small area of the abdomen, was performed and, additionally, informed consent from the horses’ owners was obtained before the initiation of the study. 

### 2.2. Hair Samples 

Hair samples were collected every two months (60.1 ± 7.3 days; mean ± SD) from December 2018 to November 2019. Therefore, six samples (*n* = 6) per individual were collected during the study period (samples were chronologically identified as P1 to P6, corresponding, respectively, to late autumn–early winter, winter, spring, late spring–early summer, summer, and autumn). Samples were collected using the “shave-reshave” method, consisting of shaving the selected area at the beginning of the study and re-shaving the regrown hair in the same area after the time of interest has ended [27]. Specifically, a 7 × 7 cm area was caudally shaved to the sternum using an electric clipper as close as possible to the skin (0.5 ± 0.5 mm). While the sampling procedure was performed by a veterinarian (S.O.-M.), an assistant was holding a piece of paper under the electric clipper to collect the hair that was being shaved. The animals tolerated the shaving process well and no fear reactions were noticed during the sampling procedure. The electric clipper was cleaned between individuals with 70% ethanol to avoid cross-contamination. Body location was selected to prevent potential injuries with the mount, to avoid visible aesthetic defects in the horses, and to avoid faecal and urine contamination [27]. Samples were stored in individually identified paper envelopes at room temperature until sample processing.

### 2.3. Hair Steroid Extraction 

For the hair steroid extraction, a previous modified and validated protocol from our laboratory was used [20]. First, 250 mg of hair was weighed using a precision scale. Each hair sample was washed by adding 2.5 mL of isopropanol (2-propanol 99.5%, Scharlau, Barcelona, Spain) and vortexed at 145× *g* for 2.5 min to remove external steroid sources. Then, the supernatant was separated by decantation. The samples were washed 3 times. Hair samples were left at room temperature for approximately 36 h to dry completely. The dried hair was milled using a ball mill (MM200, Retsch, Haan, Germany). After that, 50 mg of the milled hair was mixed with 1.5 mL of pure methanol and incubated in an orbital shaker at 36 °C and 100 rpm (G24 Environmental Incubator Shaker; New Brunswick Scientific Co. Inc., Edison, NJ, USA) 18 h for the steroid extraction. Samples were then centrifuged at 9500× *g* for 5 min, and 0.750 mL of the supernatant were transferred into a new micro-tube until complete evaporation. Once the process was completed, the dried extracts were reconstituted with 0.2 mL enzyme immunoassay (EIA) buffer provided by the commercial kit (see below) and stored at −20 °C until assay analysis.

### 2.4. Hormone Detection and Biochemical Validation Tests

Enzyme immunoassay kits (Neogen Corporation©, Ayr, UK) were used to quantify hair cortisol and testosterone concentrations (HCC and HTC, respectively). According to the manufacturer, the cross-reactivity of the EIA cortisol antibody with other steroids was prednisolone 47.4%, cortisone 15.7%, 11-deoxycortisol 15.0%, prednisone 7.83%, corticosterone 4.81%, 6β-hydroxycortisol 1.37%, 17-hydroxyprogesterone 1.36%, deoxycorticosterone 0.94%. Steroids with a cross-reactivity <0.06% are not presented. For the testosterone EIA kit, the cross-reactivity was androstenedione 0.86%, bolandiol 0.86%, testosterone enanthate 0.13%, estriol 0.10%, testosterone benzoate 0.10%, estradiol 0.05%, dehydroepiandrosterone 0.04%, testosterone propionate 0.04%, deoxycorticosterone 0.03%, testosterone 17ß-cypionate 0.02%. In this case, steroids with a cross-reactivity <0.02% are not presented. For DHEA-S, an enzyme immunoassay kit (IBL International GmbH©, Hamburg, Germany) was used. According to the manufacturer, the cross-reactivity of the EIA DHEA-S antibody was androsterone sulphate 5.67%, estrone 2.62%, testosterone 2.13%, progesterone 0.93% and, 17-α-Hydroxyprogesteronsulfat 0.13%. Substances with a cross-reactivity <0.01% are not presented. The criteria of precision, specificity, accuracy and sensitivity were followed to carry out the biochemical validation of the assays [32].

The validation tests were performed using a constituted pool made with 60 μL from each sample of the study. The precision for C, T and DHEA-S was assessed by calculating intra-assay coefficients of variation from all duplicated samples analysed. For cortisol, the linearity of the dilution was assessed using 1:1, 1:2, 1:5, 1:10 dilutions of the pool with the EIA buffer. The spike-and-recovery test was calculated by adding to 100, 75 and 25 μL of the pool, volumes of 25, 75 and 100 μL of three standard cortisol concentrations provided by the EIA kit (0.2, 0.4 and 1 ng/mL), respectively. For testosterone, the specificity of the test was assessed by calculating the linearity of the dilution using 1:10, 1:20, 1:40, 1:100 dilutions of the pool with the EIA buffer. Accuracy was assessed through the spike-and-recovery test, calculated by adding to 100, 75 and 25 μL of the pool, volumes of 25, 75 and 100 μL of three standard testosterone concentrations provided by the EIA kit (0.02, 0.04 and 0.08 ng/mL). Finally, for DHEA-S, the same procedure as followed for C and T was carried out. The linearity of the dilution was assessed using 1:1, 1:2, 1:4, and 1:6 dilutions of the pool with the EIA buffer. The spike-and-recovery test was calculated by adding to 100, 75 and 25 μL of the pool, volumes of 25, 75 and 100 μL of three standard cortisol concentrations provided by the EIA kit (0.3, 0.9 and 2.7 ng/mL), respectively. Finally, the sensitivity for C, T and DHEA-S was given by the smallest amount of hormonal concentration detected.

### 2.5. Statistical Analysis

Data were analysed using RStudio software (R version 3.4.4) and, graphically represented using GraphPad Prism software (version 8.0.2). For the biochemical validation, Pearson’s Product Moment correlation was used to evaluate the relationship between the expected and obtained values from serial dilutions of the pool with EIA buffer of C, T, and DHEA-S, respectively. The normality of the data was evaluated using a Shapiro–Wilk test and concentrations were log10 transformed, when necessary, to achieve the normal distribution. A Linear Model was used to evaluate the effect of seasonality on hair C, T and DHEA-S concentrations. The period of the year (P1-P6) was considered as fixed factor and the individual as a block. A post hoc test over periods was performed using Tukey’s multiple comparison test. For the assessment of C/DHEA-S and T/C ratios, the same model as exposed before was computed. A Tukey multiple comparison test was then performed. Finally, the relationships between hormones (C, T and DHEA-S) were explored using Pearson’s Product Moment correlation. Data are presented as mean ± SEM unless otherwise stated. The significance level in all data was set at *p* < 0.05.

## 3. Results

### 3.1. Biochemical Validation of the Enzyme Immunoassay

The intra-assay CV was 8.1% for cortisol, 4.9% for testosterone and 3.7% for DHEA-S. The dilution test revealed a significant correlation between the observed and the theoretical values for C (Pearson’s correlation; r = 0.99, *p* < 0.01), T (Pearson’s correlation; r = 0.99, *p* < 0.01) and DHEA-S (Pearson’s correlation; r = 0.99, *p* < 0.01) concentrations. In the spike-and-recovery test, the pool spiked with the hormone standards presented a mean recovery percentage of 113.6 ± 25.1% (mean ± SD) for C, 99.7 ± 2.2% (mean ± SD) for T, and 122.9± 15.2% (mean ± SD) for DHEA-S. Sensitivity was 0.013, 0.002, 0.017 ng/mL for C, T and DHEA-S, respectively.

### 3.2. Seasonal Effect on C, T and DHEA-S Concentrations

C, T and DHEA-S were significantly affected by season (*p* < 0.05) (Figure 1). HCC, from late spring–early summer (P4) (5.5 ± 0.6 pg cortisol/mg hair) and summer (P5) (5.6 ± 0.7 pg cortisol/mg hair) were significantly higher (*p* < 0.05) compared to late autumn–early winter (P1) (3.0 ± 0.1 pg cortisol/mg hair) and winter (P2) (3.8 ± 0.4 pg cortisol/mg hair). HTC were higher (*p* < 0.05) in late autumn–early winter (P1) (3.1 ± 0.4 pg testosterone/mg hair) and autumn (P6) (3.0 ± 0.2 pg testosterone/mg hair) when compared to the spring concentrations (P3) (1.9 ± 0.2 pg testosterone/mg hair). DHEA-S from late autumn–early winter (P1) (1.5 ± 0.06 pg DHEA-S/mg hair) was significantly higher (*p* < 0.05) than those concentrations obtained from winter (P2) (1.2 ± 0.07 pg DHEA-S/mg hair).

### 3.3. Seasonal Variations of C/DHEA and T/C Ratios

Both C/DHEA and T/C ratios were significantly affected by season (*p* < 0.01) (Figure 2). The C/DHEA ratio was significantly higher in summer (P5) (4.8 ± 0.7) compared to autumn (P6) (2.8 ± 0.2) and late autumn–early winter (P1) (2.0 ± 0.1). T/C ratio obtained from late autumn–early winter (P1) (0.9 ± 0.09) and autumn (P6) (0.8 ± 0.07) were higher to those C/T ratios obtained from P2 (0.6 ± 0.05), P3 (0.5 ± 0.04), P4 (0.56 ± 0.05), P5 (0.5 ± 0.07).

### 3.4. Hormone Correlations

The Pearson’s product moment correlation coefficient revealed a positive correlation between HTC and HCC (r = 0.43; *p* < 0.05). No significant correlations between T and DHEA-S or C and DHEA-S were found (*p* > 0.05).

## 4. Discussion

The use of hair to assess cortisol concentrations has become increasingly popular in many mammal species [13], but the analysis of sexual hormones and other steroids similar to DHEA-S has been less employed. To the authors’ knowledge, this is the first study that monitors the seasonal variations in T, DHEA-S, C/DHEA-S and T/C ratios in horse hair. The individuals of this study were all the same breed and sex (stallions), had similar age and same coat colour (black), and were housed in the same stable under similar management and nutritional conditions. All these conditions were selected to reduce some of the potential sources of variation in steroid hormone concentrations [13,18]. Following our findings, we detected that C, T and DHEA-S were significantly affected by seasonality, with the highest values of C during summer and the lowest values of T during spring. DHEA-S concentrations were increased in late autumn–early winter compared to winter.

Seasonal variations in HCC have been reported previously, suggesting higher HCC in autumn [17] and summer [27]. In accordance, the present results based on the sample analysis across seasons, suggests seasonal differences in HCC. We detected a progressive increase in HCC from winter to summer. The present results could be due to seasonal differences in the environmental and physical (e.g., exercise) conditions affecting long-term cortisol secretion [18]. Horses, with centuries of selection for their aptitudes, still face to potential anthropological stressful situations [33]. Horses of the present study were habitually used in the Menorca dressage discipline. More specific, during summer, horses took part in the centuries-old patron saint celebrations held every summer in Menorca (Spain). The participation of these animals into the celebrations implied a significant increase in the amount of exercise compared to the other seasons. Previous studies on humans [34] showed how exhaustive training over a period increased steroid hormone concentrations in the hair. On the other hand, previous literature has suggested exercise in horses as a potential stressor [34,35,36]. Even blood and saliva reflects acute stressful events; the repeated activation of the HPA axis in response to repeated stressful events could result in elevated hair cortisol concentrations [19,37]. Regarding our results, it could be observed that hair cortisol concentrations were lower in cold periods (P1 and P2) than in other seasonal periods, coinciding with a lower exercise rate of the studied horses during cold weather. The present results would suggest a possible relation between exercise and higher hair cortisol concentrations. On the other hand, environmental conditions would play an important role in hormonal secretion [18]. Some of the environmental factors proposed are temperature, humidity, daylight duration and weather conditions in general, but also there is a potential influence of management or individual factors in relation to the environment, such as nutrition, behaviour and metabolism [13,18]. The ambient temperature outside of the thermoneutral zone has been postulated as one of the main climatic stressors for horses [35]. To date, as far as we know, there is no literature available on how environmental factors, such as temperature, humidity or daylight duration, affect HCC in the adult horse. However, these factors have been studied in foals during the perinatal period, and no significant effect on the hair cortisol concentrations measured at birth or at 30 days of age was found [36]. In a previous study carried out in our laboratory [27], it was observed that, variations in HCC for horses located in the same climatic region as our animals were similar to those stated in the present study. The similarities in both studies reinforce the hypothesis of a marked seasonality reflected in HCC in stallions located in the Western Mediterranean region.

Seasonal variations in horse testosterone levels have been previously reported in blood [30,38], and faeces [11], although there is a limited evidence of how testosterone changes seasonally. In the horse, long days stimulate the gonadal function reaching the maximal activity in spring and summer [30,39]. Contrary to expectation, we did not detect higher HTC during spring or summer. Our results showed a lower HTC in spring (P3) compared to autumn and early winter (P1 and P6). Although we should be cautious when comparing studies using different matrixes and methodologies, our results are not in accordance with previous results [11,30], which detected higher testosterone concentrations during the breeding season in blood and faeces. The stallions from this study, even though all of them were sexually mature [39,40], were not used to breed during the study. The housing conditions, individually indoors and stabled in separated boxes, did not allow for visual contact with females or geldings. These circumstances could have interfered with the generation of sexual stimuli, leading to the increase in testosterone levels during the breeding season. Additionally, domestication and breeding selection could have reduced seasonal changes in reproductive functions in horses [41], whereas in wild and preserved breeds, or horses living outdoors, the strong seasonality could be more preserved [42].

DHEA-S showed different accumulation values in hair between seasons. Different species have shown seasonal variation in their blood DHEA-S concentrations (e.g., red squirrels or birds [43,44]) attributed to territorial behaviours since DHEA-S can be rapidly converted into sexual steroid promoters, among others, of sexual conduct [43]. The scarce information available of DHEA-S seasonal variations highlights the results obtained in this study. The variations of DHEA-S, observed in horse hair, may have a protective function by counteracting cortisol’s effects, related to the presence of recurrent or chronic stressors [3,45], as has been previously suggested in horses [8], pigs [7], and cows [10,46].

The T/C ratio has been suggested to be an indicator of the general imbalance between the mutually inhibiting HPG and HPA axes [2]. The T/C ratio has been suggested as a predictor of aggressive behaviour in different mammal species [47,48,49]. Additionally, different studies in humans applied this ratio as a general indicator of the catabolic/anabolic equilibrium during physical fitness since both hormones have antagonistic metabolic functions [2]. Concerning our results, a significant decrease from the late autumn–early winter compared to winter was observed and the ratio T/C increased from summer to autumn. Additionally, HTC and HCC were positively correlated in this study. To the best of our knowledge, the present study is the first to report a seasonal correlation of HCC and HTC. Previously, Aurich et al. [30] reported a weak but positive significant correlation between blood testosterone and salivary cortisol. Mehta et al. suggested that the levels of glucocorticoids were related to the effectiveness of testosterone functions [47,48]. When cortisol levels are high, the relationships between high testosterone and sexual behaviours tend to break down [47,48,49]. Taken together, we suggest, therefore, that an environmentally driven increase in HCC may mediate a related increase in HTC in stallions, maybe in an attempt of physiologically coping with a stressful situation.

C/DHEA-S ratio showed variations among seasons. These results may indicate different interferences between both steroids when the HPA axis was activated. The C/DHEA-S ratio has been proposed as a complementary stress indicator, since DHEA-S has a wide range of opposite effects to those produced by glucocorticoids [3]. In addition, C/DHEA-S has been postulated as a potential biomarker of resilience and allostatic load [3]. Only a few studies have been carried out in ungulates, especially on cows [10,46] and horses [8], exposing how unfavourable environmental conditions and animal management lead to an increase in the C/DHEA-S ratio. Although the present study does not focus on the behavioural observations, this study provides valuable information on the importance of considering seasonality when designing studies using hair as integrative retrospective matrix, even for calculating hormone ratios.

## 5. Conclusions

To conclude, the results of this study highlight that hair C, T and DHEA concentrations in horses change significantly among seasons. While, for HCC, we report a progressive increase in concentrations from winter to summer, with the highest values of cortisol during summer; for HTC, the lowest levels of testosterone were detected during spring. DHEA-S showed a decrease during winter compared to the last period of autumn. Hormonal ratios have been successfully assessed in horse hair and we suggest them as a complementary approach to physiological stress measurements. Additionally, the ratios evaluated in this paper (T/C and C/DHEA-S) have showed seasonal variations. Finally, it is worth remarking that season should be considered as an influencing factor when analysing sexual and stress hormones in hair, as has been demonstrated in the present study.

## Figures and Tables

**Figure 1 animals-11-02202-f001:**
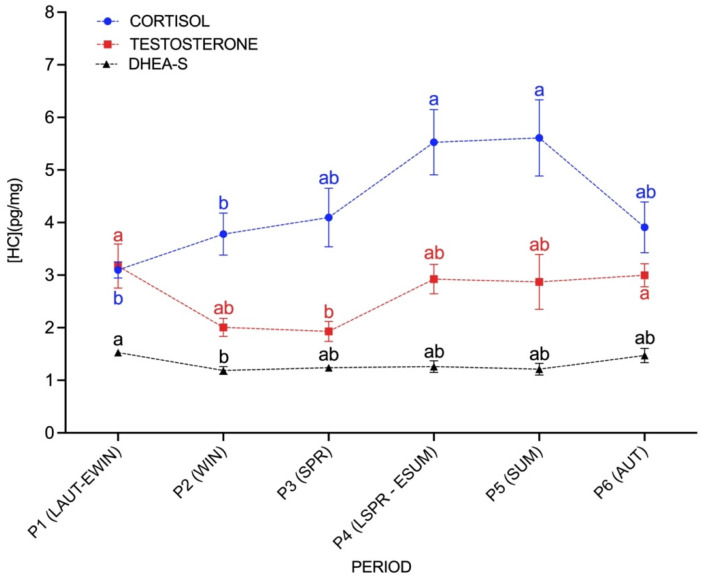
Hair C (○ blue), T (□ red) and DHEA-S (△ black) concentrations during the studied period (P1; late autumn-early winter, P2; winter, P3; spring, P4; late spring-early summer, P5; summer, P6; autumn). Data are graphically presented as mean ± SEM. HC—hormone concentrations. Different letters represent significant differences between periods (*p* < 0.05).

**Figure 2 animals-11-02202-f002:**
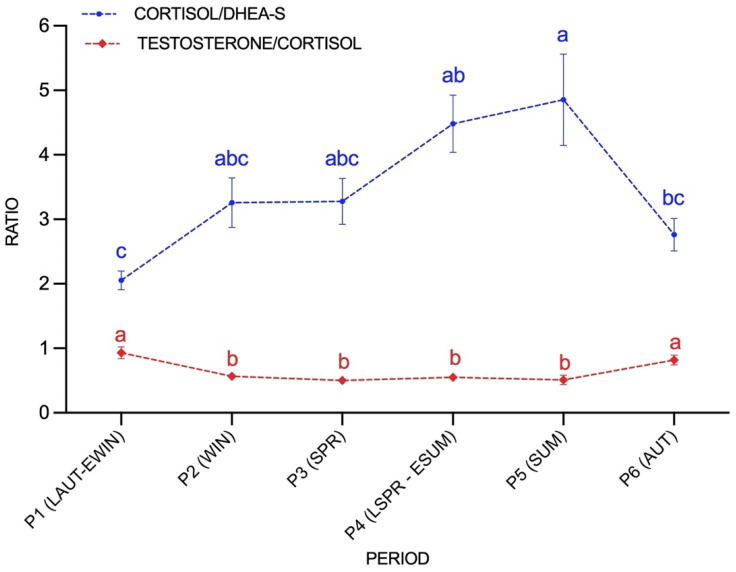
The hair C/DHEA-S (○ blue) and T/C ratio (⯁ red) variations during the studied period (P1; late autumn-early winter, P2; winter, P3; spring, P4; late spring-early summer, P5; summer, P6; autumn). Data are graphically presented as mean ± SEM. Different letters represent significant differences between periods (*p* < 0.05).

## Data Availability

Data presented in this paper have not been published or stored elsewhere, but are available on request from M.L.-B.

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
