# Peer review of "Hair Cortisol, Testosterone, Dehydroepiandrosterone Sulfate and Their Ratios in Stallions as a Retrospective Measure of Hypothalamic–Pituitary–Adrenal and Hypothalamic–Pituitary–Gonadal Axes Activity: Exploring the Influence of Seasonality"

_animals, 2021, doi:10.3390/ani11082202_

Round 1

Reviewer 1 Report

Comments and suggestions are given in the enclosed file.

Author Response

This study aims to evaluate seasonal variations in hair cortisol, testosterone and DHEA-S and to assess cortisol/DHEA-S and testosterone/cortisol ratios as a measure of stress in horses. The authors found that hair cortisol, testosterone, DHEA-S and their ratios are affected by season and recommend considering season as an influencing factor when analysing these hormones in horses’ hair.

In general, the manuscript is well written and the analytical methods and statistical evaluations used are mainly adequately described. However, more details are needed on the housing conditions and use of the animals and on the sampling method as indicated in the specific comments. Comments also concern the interpretation of the results. The findings of this study expand the knowledge about season as an influencing factor on different steroids in hair of horses.

ANSWER: We thank the reviewer for his/her comments and hope that the changes have improved the manuscript. 

Specific comments:

Summary, abstract, introduction

The authors state two objectives of the study that are obviously the same, which is to measure seasonal variations of the steroids and their ratios. An assessment of C/DHEA-S and T/C as stress indicators, however, as indicated by the second objective, cannot be provided by the results, as the variation cannot be explained by known stressful events. I suggest rephrasing the aim of the study.

RESPONSE: We have rephrased the objective of the study.

CHANGES IN THE MANUSCRIPT: The objective of the study has been changed to: "In the present study we aimed to evaluate the seasonal variations of hair C, T and DHEA-S in horses for a whole year, as well as to assess the variations between seasons of C/DHEA-S and T/C ratios."

Material and Methods

Animals and housing:

Necessary information on the housing conditions and use of the animals is needed here. Some information first comes in the discussion, for example, that the animals were housed individually indoors. Were there seasonal changes in housing conditions, e.g., outdoor access or access to pasture, sunlight exposure? For what purpose were these horses kept and could their (maybe seasonal) use be a reason for the seasonal variation in hormone levels? The colour of hair samples (black) should already be given here.

RESPONSE: All the questions and comments suggested by the reviewer have been addressed and included in the manuscript.

CHANGES IN THE MANUSCRIPT: Ten Pure Breed Menorca stallions, a black coated and autochthonous breed from the island of Menorca (Spain), were included in the study. The animals aged from eight to twelve years old. All individuals were healthy, and no symptoms of illness were detected from at least one month before the beginning of the study. The body condition score (BCS) was evaluated using the 0 (emaciated) to 5 (extremely fat) BCS scale. For all the stallions evaluated in the present study, BCS was 3 (moderate to good body condition). Additionally, a daily observation of the animals was done by the owners and a general physical examination was performed the sampling days by a veterinarian (S.O.-M). Horses were housed in Centro Ecuestre Equimar (Maó, Menorca, Spain) in individual indoor stables of 10.5 m2 with no cooling or heating elements during the different seasons. No changes in facility and housing conditions were carried out during the whole study. Individuals were fed with a similar commercial horse diet 3 times per day with ad libitum access to hay and running water. The horses from the study were all destinated to perform the same type of equestrian discipline, specifically horses were habitually used to Menorca dressage.

Lines 119-123: According to the abstract, the shave-reshave approach was used and thus the procedure must be described here. Were hairs collected by hand, which seems difficult as reshaved hair are quite short. Please describe the sampling more in detail. How did the animals tolerate the shaving in this region?

RESPONSE: According to the reviewer, some sampling details have been added in the manuscript.

CHANGES IN THE MANUSCRIPT: Samples were collected using the “shave - re-shave” method, consisting in shaving the selected area at the beginning of the study, and re-shaving the regrown hair in the same area after the time of interest has ended (Gardela et al., 2020). Specifically, a 7 × 7 cm area was shaved caudally to the sternum using an electric clipper as close as possible to the skin (0.5 ± 0.5 mm). While the procedure was performed by a veterinarian (S.O.-M.), an assistant was holding a piece of paper under the electric clipper to collect the hair that was being shaved. The animals tolerated well the shaving process and no fear reactions were noticed during the sampling procedure. The electric clipper was cleaned every time between individuals with 70% ethanol to avoid cross-contamination. The body location was selected to prevent potential injuries with the mount, to avoid visible aesthetic defects in the horses, and to avoid faecal and urine contamination. Samples were stored in individually identified paper envelopes at room temperature until sample processing.

Line 137: should read „Retsch“

RESPONSE: The change has been added.

The analytical methods and their validation are adequately described. I only miss data on the sensitivity and maybe inter-assay CV of the assays.

RESPONSE: The sensitivity has been added in the manuscript at the results section (part 3.1 “Biochemical validation of the enzyme immunoassay”). Additionally, a sentence at the M&M section (L) has been added to explain it. The inter-assay CV are not applicable because the whole number of samples were analysed in a single assay per hormone to avoid inter-assay variations.

Results

Lines 203-211: Concentrations are given in ng/mg but are certainly pg/mg.

RESPONSE: We thank the reviewer for the comment. The units of the concentrations in the results section have been corrected to pg/mg.

Figure 1 and 2: It is difficult to figure out which sample (P1-P6) belongs to which season. I strongly recommend using season or months on the x-axis.

CHANGES IN THE MANUSCRIPT:

  • Section 2.2: A sentence has been added relating the season to the corresponding study period.
  • Figure 1 and 2: The season corresponding to each period have been added in both Figure 1 and Figure 2 “X” axes.

Figure 2: The ratio T/C could be coloured in red since T is the numerator, comparatively similar to the C/DHEA-S ratio in blue.

RESPONSE:  The colour of the ratios has been changed accordingly.

Discussion

Line 237-238:  DHEA-S was only higher in autumn compared with winter but not to the other periods. Please correct.

RESPONSE: The reviewer is right, and we have performed the modification.

CHANGES IN THE MANUSCRIPT: the sentence has been changed by: DHEA-S concentrations were increased in late autumn- early winter compared to winter.

Lines 239-253: It is clear that seasonality is a multifactorial influencing variable. But could there be assumption from the specific housing and environmental conditions or the use of the animals what factors may have had a major influence in this study?

RESPONSE: According to the reviewer’s question we have added a paragraph discussing the influence of both exercise and environmental factors and their potential relevance on hair hormonal levels. Both type of factors may have a major influence on hormone concentrations.

Lines 270-271: I think this sentence makes no sense and should be rephrased since the pattern of DHEA levels is not necessarily different from T but particularly different from C.

RESPONSE: The suggested change has been added into the manuscript.

CHANGES IN THE MANUSCRIPT: the sentence has been changed to: DHEA-S showed different values in hair between seasons.

The partially antagonistic effects of the steroids is described in the discussion and forms the rationale for the two ratios. Thus, from the results, there seems to be an opposite seasonal time course of DHEA-S compared to C. In addition, there is a noticeable parallel time course of DHEA-S with T. Since DHEA-S can be transformed to T via DHEA, could this cause the similar pattern? The authors could discuss their findings in a bit more detail with regard to the physiological relationships of these steroids. In this context, it could have been useful to measure correlations.

RESPONSE: We thank you the reviewer for his/her comment. This topic (DHEA-S as a precursor of T at the adrenal gland and therefore plausible similar time course) has been included into the Discussion section.

Lines 289-290: copying? I suppose the phrase should read like „coping with a stressful situation“

RESPONSE: The change has been added.

Reviewer 2 Report

Thank you for the opportunity to review your manuscript. I enjoyed reading it and found the work interesting. Please see below my comments and suggestions for improving your manuscript.

Some grammar issues and lack of clarity is impacting on the readability and comprehension of your manuscript. Double spaces and some overly long sentences throughout disrupt the flow of the writing.

Opening sentence of the simple summary is a little confusing, I suggest rewording to clarify.

Abstract needs a little clarification in L37 it is unclear what “variation among seasons” refers to, is it between seasons or within seasons?

Introduction requires some grammar revising throughout, some examples are listed below. Good background and justification for the study.

Methods are more clearly written and well laid out. Could have more detail on the nutrition and management of the animals where relevant. For example, it would be useful to indicate whether the animals were stabled or in paddocks (stabling is briefly referred to in the discussion), and what form of exercise was conducted during the study period as intense exercise can impact on hair cortisol (Gerber et al. 2012; Skoluda et al. 2012). Did all of the study animals have the same exercise regimen in terms of regularity and intensity? Type of nutrition, number of feeds per day etc may also influence the measures if the animals did not receive equivalent nutrition and so this needs to be noted. It is not clear from the methods which season each period represents, I recommend clarifying this in section 2.2. Statistical analysis appears appropriate for the data and clearly outlined.

Results are nicely presented, inconsistent spacing throughout hinders the flow and appearance for the reader. Figure labels should show season as well as study period.

Discussion section is a little too brief and lacks depth. There is some discussion about unknown impacts of environmental factors on these measures on the horse, it would be appropriate to discuss examples from other species briefly to assist interpretation of the data. It would also be appropriate to discuss the potential influence of exercise (or lack of exercise) on hormone production and whether it may have influenced these results.

Overall, an interesting and useful study. The manuscript would benefit from revision, particularly the introduction and discussion sections with a view to improving clarity and readability. Improving the methods will help the reader orient to season and conditions of your study. I suggest expanding the discussion to provide the reader with a more thorough analysis of the implications of your findings, recent work in fur seals by Otsuki et al. (2021) may be of use. I recommend including discussion and interpretation of influences such as behaviour and exercise and how they may be implicated in seasonal variations.

I look forward to reading your revised manuscript.

L24 need period at end of sentence

L25 remove “the”

L37 reword sentence, “variations among seasons” is unclear, do the authors mean there was variation between seasons or within season?

L44 Needs rewording, “follow up” implies that these investigations are following something but no reference as to what is being followed up.

L53 Remove double space, add “it” after “since”.

L54 Replace “action” with “actions”

L64 Remove double space.

L163 Remove double space

L219 change “on” to “in” or “during” summer for clarity.

L232-233 Coat colour and stabling information should be included in the methods.

L249 Change “now” to “know”.

L265-267 Consider rewording, this sentence is a little awkward.

L275-277 Consider rewording, it is a little vague and unclear.

L287-290 Consider rewording.

L291 Change “furtherly” to “further”.

Gerber, M, Brand, S, Lindwall, M, Elliot, C, Kalak, N, Herrmann, C, Pühse, U, Jonsdottir, IH (2012) Concerns regarding hair cortisol as a biomarker of chronic stress in exercise and sport science. Journal of sports science & medicine 11, 571-581.

Otsuki, M, Horimoto, T, Kobayashi, M, Morita, Y, Ijiri, S, Mitani, Y (2021) Testosterone levels in hair of free-ranging male northern fur seals (Callorhinus ursinus) in relation to sampling month, age class and spermatogenesis. Conserv Physiol 9, coab031.

Skoluda, N, Dettenborn, L, Stalder, T, Kirschbaum, C (2012) Elevated hair cortisol concentrations in endurance athletes. Psychoneuroendocrinology 37, 611-7.

Author Response

Thank you for the opportunity to review your manuscript. I enjoyed reading it and found the work interesting. Please see below my comments and suggestions for improving your manuscript.

RESPONSE: We thank the reviewer for his/her comments and hope that the included changes have improved the manuscript.

Some grammar issues and lack of clarity is impacting on the readability and comprehension of your manuscript. Double spaces and some overly long sentences throughout disrupt the flow of the writing.

RESPONSE: Double spaces have been checked and removed from the text. Additionally, grammar has been revised to improve the clarity and readability of the paper.

Opening sentence of the simple summary is a little confusing, I suggest rewording to clarify.

RESPONSE: We agree with the reviewer and the correspondent change has been added.

CHANGES IN THE MANUSCRIPT: L13-14 and L49-50  The assessment of a horse’s endocrine status can provide valuable information of its welfare, sexual and health conditions.

Abstract needs a little clarification in L37 it is unclear what “variation among seasons” refers to, is it between seasons or within seasons?

RESPONSE: A clarification has been added.

CHANGES IN THE MANUSCRIPT: “variations among seasons” has been changed by “variations between seasons”.

Introduction requires some grammar revising throughout, some examples are listed below. Good background and justification for the study.

RESPONSE: We thank the reviewer for his/her comments and hope that the changes will help to fix the grammar issues. All the examples listed have been changed in the manuscript.

Methods are more clearly written and well laid out. Could have more detail on the nutrition and management of the animals where relevant. For example, it would be useful to indicate whether the animals were stabled or in paddocks (stabling is briefly referred to in the discussion), and what form of exercise was conducted during the study period as intense exercise can impact on hair cortisol (Gerber et al. 2012; Skoluda et al. 2012). Did all of the study animals have the same exercise regimen in terms of regularity and intensity?

RESPONSE: We appreciate the reviewer’s comment. This issue has been addressed with changes in the manuscript.

CHANGES IN THE MANUSCRIPT: L113-119: Horses were habitually housed in Centro Ecuestre Equimar (Maó, Menorca, Spain) in individual indoor stables of 10.5 m2 with no cooling or heating elements during the different seasons. No changes in facility and housing conditions were carried out during the whole study. Individuals were fed with a similar commercial horse diet 3 times per day with ad libitum access to hay and running water. The horses from the study were all destinated to perform the same type of equestrian discipline, specifically horses were habitually used to Menorca dressage discipline.

Type of nutrition, number of feeds per day etc may also influence the measures if the animals did not receive equivalent nutrition and so this needs to be noted.

RESPONSE: We have emphasized that horses were all fed with a commercial diet three times per day with ad libitum access to hay and running water.

It is not clear from the methods which season each period represents, I recommend clarifying this in section 2.2. Statistical analysis appears appropriate for the data and clearly outlined.

RESPONSE: We agree with the reviewer.

CHANGES IN THE MANUSCRIPT:

  • Section 2.2: A sentence has been added relating the season to the corresponding study period.
  • Figure 1 and 2: The season corresponding to each period have been added in both Figure 1 and Figure 2 “X” axes.

Results are nicely presented, inconsistent spacing throughout hinders the flow and appearance for the reader. Figure labels should show season as well as study period.

RESPONSE: Figure labels changed accordingly.

CHANGES IN THE MANUSCRIPT: Both X and Y axes for Figures 1 and 2 have been changed according to the suggestion made by the reviewer.

Discussion section is a little too brief and lacks depth. There is some discussion about unknown impacts of environmental factors on these measures on the horse, it would be appropriate to discuss examples from other species briefly to assist interpretation of the data. It would also be appropriate to discuss the potential influence of exercise (or lack of exercise) on hormone production and whether it may have influenced these results. Overall, an interesting and useful study. The manuscript would benefit from revision, particularly the introduction and discussion sections with a view to improving clarity and readability. Improving the methods will help the reader orient to season and conditions of your study. I suggest expanding the discussion to provide the reader with a more thorough analysis of the implications of your findings, recent work in fur seals by Otsuki et al. (2021) may be of use. I recommend including discussion and interpretation of influences such as behaviour and exercise and how they may be implicated in seasonal variations.

RESPONSE: We thank the reviewer for these comments. According to the reviewer’s comment, we have added a paragraph discussing the influence of both exercise and environmental factors and their potential relevance on hormone production and hair hormonal levels. Both type of factors may have a major influence on hormone concentrations.

I look forward to reading your revised manuscript.

L24 need period at end of sentence

RESPONSE: Revised and added as requested.

L25 remove “the”

RESPONSE: Revised and changed as requested.

L37 reword sentence, “variations among seasons” is unclear, do the authors mean there was variation between seasons or within season?

 RESPONSE: Revised and changed as requested.

L44 Needs rewording, “follow up” implies that these investigations are following something but no reference as to what is being followed up.

RESPONSE: Revised and changed by assessment.

L53 Remove double space, add “it” after “since”.

RESPONSE: Revised and added as requested.

L54 Replace “action” with “actions”

RESPONSE: Revised and added as requested.

L64 Remove double space.

RESPONSE: Revised and removed as requested.

L163 Remove double space

RESPONSE: Revised and removed as requested

L219 change “on” to “in” or “during” summer for clarity.

RESPONSE: Done

L232-233 Coat colour and stabling information should be included in the methods.

RESPONSE:  coat colour and housing conditions have been added in the M&M section as requested by the reviewer.

L249 Change “now” to “know”.

RESPONSE: Reviewed.

L265-267 Consider rewording, this sentence is a little awkward.

RESPONSE: Reviewed

L275-277 Consider rewording, it is a little vague and unclear.

RESPONSE: Reviewed

L287-290 Consider rewording.

RESPONSE: Reviewed

L291 Change “furtherly” to “further”.

RESPONSE: Reviewed

Reviewer 3 Report

The aim of the paper ID: animals – 1212148 was the evaluation of seasonality on cortisol (C), testosterone (T) concentrations for a whole year in stallions’ hair and their interrelationship. The object of the present investigation is very interesting as, with the exception of hair cortisol, to date there is no information published on the seasonal variations in hair T levels and on the biological influence of cortisol/ dehydroepiandrosterone sulfate ester and T/C ratios in hair as potential marker of stress in horses. Morevover, the simultaneous assessment of hormones, secreted by the hypothalamic-pituitary-adrenal (HPA) and hypothalamic-pituitary-gonadal (HPG) axis activation, would be a promising approach to physiologial stress measurement. The summary is adequately focused. The introduction is satisfactory, very well exposed, and the purpose of the study is very clear. The description of the materials and methods, the formulation of the horse housing, diet handling, ethical considerations and  hair sampling is precise and accurate. Hormone concentrations were quantified by enzyme immunoassay kits. Precision, specificy and accuracy were performed by biochemical validation tests. Steroid hormones are lipophilic molecole therefore they are incorporated into the hair through passive diffusion from blood. The measuement of hair C, T and DHEA-S concentrations is a good approach to quantify circulating steroid hormone levels. The proposed method is non-invasive and pain-free sample collection. Hair is a stable substrate with less variability than other biological fluids as saliva or feces and it is easily stored at room temperature during long periods. Another advantage of this sampling is that hair steroid hormone levels are not affected by circadian rithm variations. The statistical methods, for data analysis, biochemical validation, the normality of the data and  the effect of the seasonality on hair C, T and DHEA-S concentrations, for he assessment of C/DHEA-S and T/C ratios, are adequate. The statistical methods, for data analysis, biochemical validation, the normality of the data and  the effect of the seasonality on hair C, T and DHEA-S concentrations, for the assessment of C/DHEA-S and T/C ratios, are adequate. The presentation of results is  clearly concentrated on the Figures, which are neat and easy for the reader to consider. Biochemical validation of the enzyme immunoassay showed the intra-assay CV 8.1% for cortisol, 4.9% for testosterone and 3.7% for DHEA-S. The dilution test revealed a significant correlation between the observed and theoretical values for C,T and DHEA-S. Hair cortisol concentrations were
significantly higher during late spring-early summer (P4) and summer (P5)
compared to late autumn-early winter (P1) and winter (P2). Hair testosterone concentrations were significantly higher in late autumn-early winter (P1) and autumn (P6) when compared to the spring concentrations (P3). DHEA-S concentrations from late autumn-early winter (P1) were significantly higher compared to those obtained in winter (P2). Both C/DHEA and T/C ratios were significantly affected by season. The C/DHEA ratio was significantly higher on summer (P5) compared to autumn (P6) and late autumn-early winter (P1). Hair testosterone and cortisol concentrations showed a positive correlation during the studied period, with higher values on late autumn-early winter (P1) and autumn (P6). These results showed clearly that season should therefore be considered when assessing sexual and stress hormones in stallions’ hair to avoid misinterpretations since this variable can be a potential influencing factor. The discussion section is thorough and has underlined the effect of seasonality on steroid hormone levels and their interrelationhips. The complexity of the interactions between hypothalamic-pituitary-adrenal (HPA) and hypothalamic-pituitary-gonadal (HPG) axis activation and the interaction with internal factors and environmental factors such as temperature, humidity, daylight duration and weather conditions in general, but also a potential influence of management or individual factors in relation to the environment, such as nutrition, behaviour and metabolism on the levels of hair cortisol, testosterone  and dehydroepiandrosterone sulfate ester in horses have been discussed in detail. This is the first study that monitors the seasonal variations of T, DHEA-S, C/DHEA-S and T/C ratios in adult horses’ hair reducing some of the potential sources of variation in steroid hormone concentrations. In fact, horses used for this investigation were all of the same breed and sex (stallions), had similar age and same coat colour (black), and were housed in the same stable. C, T and DHEA-S levels were significantly affected by seasonality, with the highest values of C during summer and the lowest values of T during spring, in accordance with previous results reported. The ambient temperature has been postulated as one of the main climatic stressor for horses, although to date environmental factors have been studied only in foals. DHEA-S concentrations were increased in autumn compared to the other studied periods. Concerning testosterone concentrations, contrary as expected, results showed a lower HTC in spring (P3) compared to autumn and early winter (P1 and P6). In fact, in the horse, long days stimulate the gonadal function reaching the maximal activity in spring and summer, as reported previously during the breeding season using different matrixes as blood and faeces and different housing conditions. The stallions used in this study, even being all sexually mature, were not used to breed during the whole study. The housing conditions, individually indoor stabled in separated boxes, did not allow visual contact with females or geldings. These circumstances could have been interfering the generation of sexual stimuli leading to the increase of testosterone levels during the breeding season. Additionally, domestication and breeding selection could have reduced seasonal changes in reproductive functions in horses. The variations of DHEA-S, which can be rapidly converted into sexual steroids promoters, observed in horses’ hair, may contribute to play a protective function by counteracting cortisol's effects related to the presence of recurrent or chronic stressors, as it has been suggested previously in horses, pigs and cows. T/C ratio has been suggested to be an indicator of the general imbalance between the mutually inhibiting HPG and HPA axes. The T/C ratio has been suggested as a predictor of aggressive behaviour in different mammal species. Additionally, different studies in humans applied this ratio as a general indicator of the catabolic/anabolic equilibrium during physical fitness since both hormones have antagonistic metabolic functions, catabolic cortisol and aanabolic testosterone. Concerning results of this investigation, an inverse pattern between the two hormones was observed. Taking together, an environmentally driven increase of HCC may mediate in a related decrease of HTC in stallions, or viceversa, as an attempt of physiologically copying a stressful situation. C/DHEA-S ratio showed variations among seasons. These results may indicate different interferences between both steroids when the HPA axis was activated. C/DHEA-S ratio has been proposed as complementary stress indicator since DHEA-S has a wide range of opposite effects to those produced by glucocorticoids. In addition, C/DHEA-S has been postulated as a potential biomarker of resilience and allostatic load. Only a few studies on cows and horses, exposing on unfavourable environmental conditions and animals management, lead to an increase of the C/DHEA-S ratio. In order to understand the mechanisms and consequences of this relationship further studies are needed, by performing behavioural observations to complement the physiological indicators. The results obtained have been correctly commented in relation to the physiology of the horses. References are appropriately cited. In conclusion, this study clearly demonstrated that hair C, T and DHEA concentrations in horses change significantly among seasons and therefore season should be considered as an influencing factor when analysing sexual and stress hormones in stallions’ hair. Hormonal ratios have been assessed successfully in horses’ hair. This study could have a potential application in order to provide a complementary approach to physiological stress measurements. In general, the writing style and English use in the manuscript are adequate.

Author Response

ANSWER: We appreciate the accurate review of the paper. We have included some of the comments from the reviewer into the discussion section and improved the figures for more clarity. Thank you very much for your constructive criticism.

Round 2

Reviewer 1 Report

Lines 352-354: This end of the sentence still makes no sense. Instead of "copying a stressful situation" "coping with a stressful situation" is probably meant.